# Plus-InVEST Study of the Chengdu-Chongqing Urban Agglomeration's Land-Use Change and Carbon Storage

Chaoyue Wang [1], Tingzhen Li [1], Xianhua Guo [1,*], Lilin Xia [1], Chendong Lu [2] and Chunbo Wang [1]

1. Research Institute of Three Gorges, Chongqing Three Gorges University, Chongqing 404100, China
2. Institute of Strategic Planning, Chinese Academy of Environmental Planning, Beijing 100043, China
* Correspondence: guoxianhua@sanxiau.edu.cn

**Abstract:** Based on China's "carbon neutrality" strategy, this study explores the relationship between land-use/cover change and temporal and spatial changes of ecosystem carbon storage in urban agglomerations. Using the Plus-InVEST model, the projected spatial patterns of land use in the Chengdu-Chongqing urban agglomeration in 2030 under natural development and ecological protection scenarios were simulated and predicted, and the characteristics of carbon storage, together with its spatio-temporal dynamics, were evaluated under two scenarios. Results show that: (1) From 2000 to 2020, forests, water areas, construction areas, and unused land continued to increase, while the area of cropland and grassland decreased continuously. During the last 20 years, carbon storage in urban agglomeration showed an increasing trend, with an overall increase of $24.490 \times 10^6$ t. (2) Compared with the natural development scenario, forest land, grassland, and water area in 2030 under the ecological protection scenario exhibits a substantial change; the area of construction land is limited; and an ecological spatial effect is reflected. (3) Compared to 2020, carbon storage under natural development and ecological protection scenarios decreased by $50.001 \times 10^6$ t and $49.753 \times 10^6$ t in 2030, respectively. The stability of carbon storage under the ecological conservation scenario was significantly higher than that under the natural development scenario. Therefore, under the ecological protection scenario, as a result of the coordinated land use of Chengdu-Chongqing, the functions of various regions can be coordinated and carbon storage losses can be mitigated.

**Keywords:** carbon storage; PLUS model; InVEST model; land use; urban agglomeration

## 1. Introduction

During the last few decades, the global carbon cycle has received considerable attention due to the storage of carbon in terrestrial eco-systems [1,2]. The primary driver behind changes in carbon storage in ecological processes is variability in land-use type [3]. Carbon sequestration capacity varies considerably out all over land-use types. Ecological processes store carbon in plants and soils, which are influenced by changes in land use [4,5]. Currently, industrial development are causing a massive development of urban land areas [6]. The rise in developed land and resulting loss of natural vegetation have had a significant impact on regional carbon storage, posing a serious threat to sustainability and the supply of regional ecological processes [7]. Timely and effective assessments of regional carbon storage affected by urban agglomeration construction and development are crucial to maintain carbon storage services while enhancing other ecosystem services [8,9]. Thus, the sustainable development of urban agglomerations can be improved by providing information to enable the coordination of land use [9–11].

Land-use/cover change (LUCC) impacts carbon storage using field investigations and modeling [12]. This is a complex process that has both spatial and temporal aspects [13]. Several models, including Conversion of Land Use and its Effects at Small Region Extent (CLUE-S) [14] and the Land Use Scenario Dynamics (LUSD) model [15], are suitable for assessing urban areas. Additionally, the Cellular Automata-Markov (CA-Markov) model

has gained popularity for simulating LUCC in a variety of situations and produces accurate results [16,17]. In one study, Liang et al. (2021) combined a CA-Markov model with an InVEST model to assess the impact of land-use change on global key ecological carbon stocks [18,19]. In addition, the Simulation of Future Land Use (FLUS) model has been applied in scenario analyses to a certain extent due to its different operation mode relative to the CA model [20]. A combination of FLUS and the InVEST model was used by Deng et al. [21], Liu et al. [22], and Gao [23] to examine the relationship between future land use and carbon storage in the future. However, utilizing the land expansion analysis technique, a patch-generated land-use change simulation tool refers to a network data may more accurately assess the reasons behind diverse land-use changes (LEAS). The model PLUS includes a multiseed growth mechanism (CARS) that better simulates patch-level changes across multiple land uses, enabling the appraisal of multiple land-use types [24]. Depending on the use of the LEAS and CARS modules [25], under several anticipated future scenarios, the PLUS model can also provide an accurate assessment of how urban expansion affects carbon storage in land ecosystems.

In the upper reaches of the Yangtze River, the Chengdu-Chongqing urban agglomeration is located in an ecological barrier area. In response to the rapid loss of cultivated land resources due to the expansion of urban and rural construction land as well as occupying ecological land, this urban agglomeration faces severe challenges when it comes to production, living, and ecological spaces [26,27]. It is thus very important to explore evolution, simulation, and scenario prediction in this region. This paper examines the potential impacts of future urban agglomeration development on regional carbon storage. Our study examines land change from a territorial spatial evolution perspective taking into account the impacts of natural, social, economic, and transportation factors. We quantitatively simulated regional land-use change in urban agglomerations between 2020 and 2030 as well as determined whether different spatial regulation scenarios might have a significant impact on regional carbon storage in Chengdu-Chongqing. The planning space should be used for a variety of spatial regulation purposes. The objective of the study is to explore urban agglomerations effectively and alleviate known contradictions between urban development and environmental conservation by attempting to explore urban agglomeration development and alleviate the known contradictions between urban development and environmental protection.

## 2. Materials and Methods

### 2.1. Study Area

The Chengdu-Chongqing urban agglomeration, which has its centers in Chengdu and Chongqing, is a crucial platform for the growth of the western area and a vital ally for the Yangtze River Economic Belt and an important area for China to promote new-type urbanization. The agglomeration includes 15 cities in Sichuan province and 29 districts (counties) in Chongqing. As shown in Figure 1, the permanent population in this area was 97,709,900 in 2021, accounting for 6.8% of the national population, and its economic aggregate in the same year accounted for 6.5% of the national total. Chengdu and Chongqing influence the surrounding areas by virtue of their relative economic strength. The inflow of Chongqing population into Chengdu accounted for 4.32%, while the inflow of Chengdu population into Chongqing accounted for 7.68%, demonstrating a "dual flow" development of both cash flow and traffic flow.

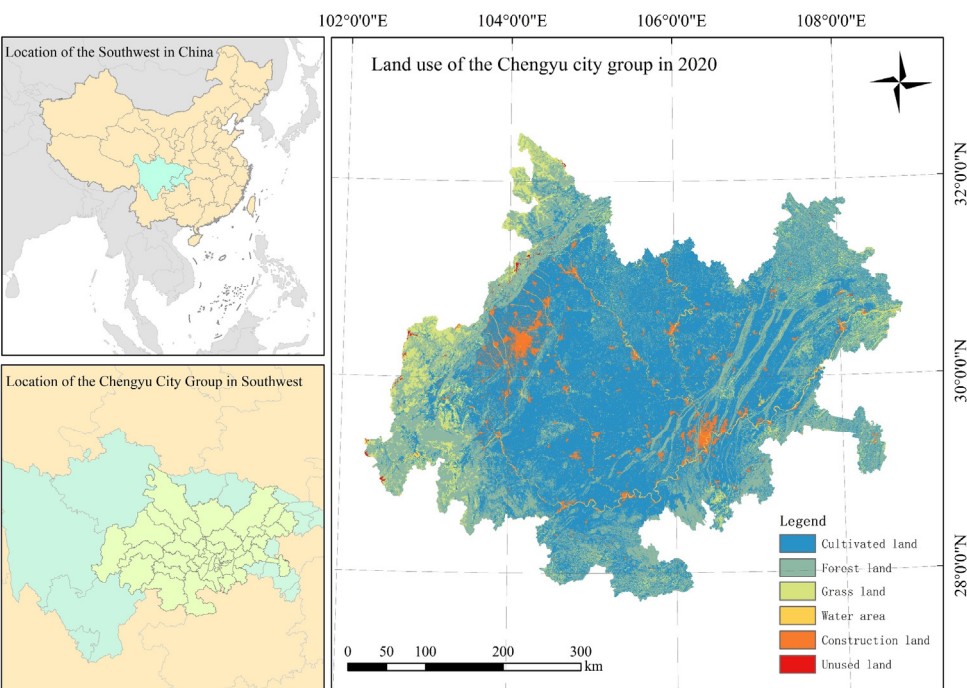

**Figure 1.** Map of the study area.

## 2.2. Data Acquisition and Processing

For the simulation of land use and carbon storage in this article, the following data used here are briefly described, including the land use and carbon storage simulation data used in this paper. Given the accessibility of remotely sensed data, Landsat pictures with a pixel size of 30 m for the years 2000, 2010, and 2020 (containing Thematic Mapper, Enhanced Thematic Mapper, and Operational Land Imager) were obtained from the Geographic Information Cloud site (http://www.gscloud.cn, accessed on 6 June 2021) to categorize land use and cover. Land-use types were divided into 6 categories and 25 subcategories [28]. Resource and Environment Science and Data Center (http://www.resdc.cn/, accessed on 12 June 2021) provided the digital elevation model (DEM), slope, gross domestic product (GDP), and population data. DEM and slope data were processed at a spatial resolution of 30 m, while GDP and population data were processed at a spatial resolution of 1 km. Point of Interesting(POI), river, and night light data were also obtained from RESDC. Our road network data was derived from OpenStreetMap (https://www.openstreetmap.org/, accessed on 2 July 2021). With a spatial resolution of 100 m, annual mean temperatures and annual mean precipitation data were collected from World Clim (https://www.worldclim.org/, accessed on 16 August 2021) [29].

In ArcGIS 10.8 (ArcGIS 10 series created by Esri (Redlands, CA, USA)), a unified spatial resolution of 100 m × 100 m was set for all of the above-mentioned spatial data, adopting the Albers geographic coordinate system. Driving factors can be divided into the following four categories (Figure 2): terrain, climate, location, and social and economic. Aspect, slope, and elevation are topographic factors. Temperature and precipitation are examples of climatic factors. Location factors include distance to rivers, roads at all levels, and schools. Distances were calculated using the ArcGIS Euclidean distance tool. Socioeconomic factors include GDP per capita, population density, and nighttime lighting conditions [30,31].

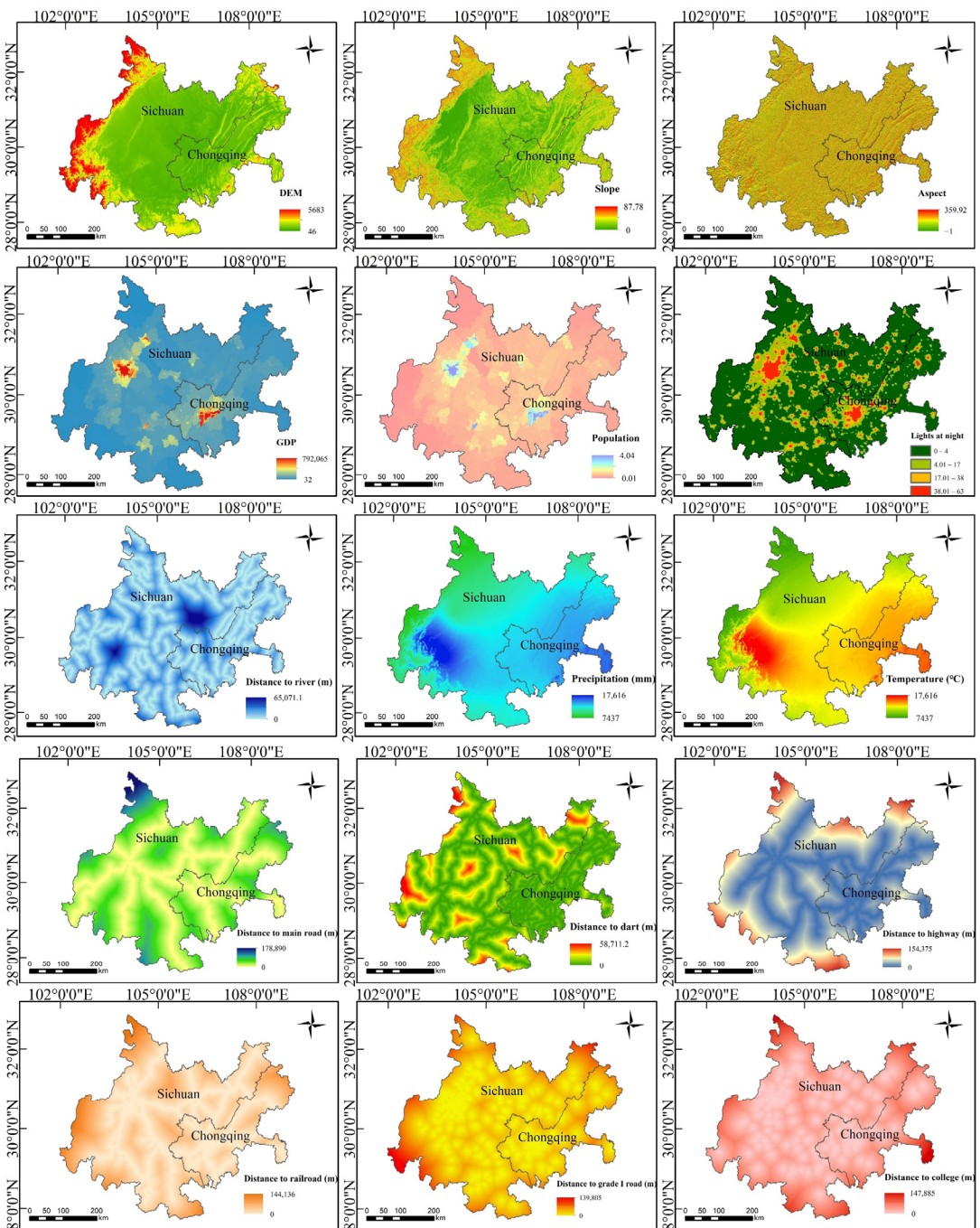

**Figure 2.** Main drivers behind land usage in the urban agglomeration of Chengdu and Chongqing.

*2.3. Research Methods*

2.3.1. PLUS Model

A model of land-use change called PLUS is based on patches of grid data. The modeling can replicate diverse changes in land use and properly characterize them at the component level. Two modules make up the PLUS model: Based on several random patch seeds, CARS is a CA model and LEAS a land extension analysis technique [24]. The LEAS module may harvest and sampling land expansion between two periods of land-use change, utilizing the random forest algorithm to mine and determine the likelihood that different land uses will emerge as well as the percentage of driving variables that each land use will contribute. The CARS module simulates autonomous plaque production under the constraint of development probability by combining the mechanisms of random seed

formation and cutoff decline. Based on the PLUS model, the LEAS module was used to analyze land expansion from 2000 to 2020. Subsequently, the demand for land use by 2030 was estimated using the Markov chain algorithm. Under two distinct 2030 development scenarios, the CARS module was used to simulate and predict land-use changes.

The development risk surfaces $P_{i,k}^{d=1}$ for land use and overall likelihood $OP_{i,k}^{d=1,t}$ may be calculated using the Monte Carlo method when $\Omega_{i,k}^t$ is 0 [32], as follows:

$$OP_{i,k}^{d=1,t} = \begin{cases} P_{i,k}^{d=1} \times (r \times \mu_k) \times D_k^t \text{ if } \Omega_{i,k}^t = 0 \text{ and } r < P_{i,k}^{d=1} \\ P_{i,k}^{d=1} \times \Omega_{i,k}^t \times D_k^t \text{ all others} \end{cases} \tag{1}$$

where r varies between 0 and 1; the threshold for creating new land-use patchwork for land-use type k is represented by $\mu_k$, which the user chooses. $\Omega_{i,k}^t$ is the percentage of land-use k that makes up the area around cell i; and $D_k^t$ denotes the gap between present and future land-use demands at iteration t. $\tau$ is used to evaluate the nominated land use c, which is selected by the roulette wheel, if land-use c is more common than land-use k:

$$\text{If } \sum_{k=1}^{N} \left| G_c^{t-1} \right| \sum_{k=1}^{N} \left| G_c^t \right| < \text{ Step Then, } l = 1 + 1 \tag{2}$$

$$\begin{cases} \text{Change } P_{i,c}^{d=1} > \tau \text{ and } TM_{k,c} = 1 \\ \text{No change } P_{i,c}^{d=1} \leq \tau \text{ or } TM_{k,c} = 0 \end{cases} \tau = \delta^l \times r1 \tag{3}$$

where Step refers to the step size needed by the PLUS model to roughly represent the land-use requirement; as $\delta$ is the decay factor for $\tau$, which ranges from 0 to 1, the decay factor is set by the expert; l is the total number of decay steps, and r1 is a normal distributions stochastic variable with a mean of 1 and a range of 0 to 2. The transition matrix, $TM_{k,c}$, determines whether land-use type k may change to type c [24,33].

The interaction between various land-use types and various land-use divisions within the neighborhood is another neighborhood component [24], which can be said in the following manner:

$$\Omega_{p,k}^t = \frac{\sum_{N \times N} \text{con}\left(C_p^{t-1} = k\right)}{N \times N - 1} \times w_k \tag{4}$$

where $\Omega_{p,k}^t$ is the local effect factor for the cell p at time t and is the entire amount of cells that land type k occupied in the Moore neighborhood window of $N \times N$ in the previous iteration $t - 1$; and $w_k$ represents the neighborhood factor parameter of each land-use type. The neighborhood factor parameter ranges from 0 to 1, with a value proportional to land expansion capacity [34]. The land-use factor parameters in this paper are primarily based on current situations and future development trends of land use in the study area (Table 1).

**Table 1.** Neighborhood factor parameters.

| Land Use Type | Cultivated Land | Forest | Grassland | Water | Construction Land | Unused Land |
|---|---|---|---|---|---|---|
| Natural development neighborhood factor | 0.07 | 0.11 | 0.01 | 0.29 | 1 | 0.09 |
| Ecological protection neighborhood factor | 0.07 | 0.31 | 0.10 | 0.34 | 0.95 | 0.09 |

2.3.2. Validation of Model Accuracy

The applicability and reliability of the model for forecasting changes in land use and cover were assessed using quantifiable correctness and the kappa coefficient. The overall agreement between simulation findings and observation data is tested using the kappa value. Kappa values greater than 0.75 signify good simulation accuracy. Taking 2010 as the base period data, the paper uses the above methods to simulate 2020 land-use patterns,

then cross-checks the simulation graph of 2020 and the current situation graph of 2020. Calculating the kappa coefficients is as follows:

$$\text{Kappa} = \frac{OA_O - OA_E}{(1 - OA_E)}, OA_O = \left(\sum_{k=1}^{n} OA_{kk}\right)/N \tag{5}$$

where $OA_O$ is the classification's overall accuracy and denotes the likelihood that each random sample's simulation outcome would match the data on land use. $OA_E$ is the likelihood that the simulation's findings match the data on current land use; the number n represents the number of types of land use; N is the overall sample count; the quantity of samples that were accurately identified for the k type of land use is called $OA_{kk}$. The range of values for the kappa coefficient is −1 to 1, with a higher number indicating a more appropriate prediction.

### 2.3.3. Setting the Scene

Natural development scenario (NDS): In light of the land-use development trend between 2000 and 2020, With the Markov chain, it was possible to determine the demand for land usage in 2030 underneath the historical development trend (Table 2) [35,36]. According to historical changes, cultivated land has become grassland or construction land, so we set it to 1. Since it is unlikely to turn to other ground classes, it is set to 0. A similar situation exists for woodlands and arable lands. Despite its particularity, construction land cannot be converted to other land classes, so it is set at 0. In the past, unused land has more often migrated to OTHER land classes than to the rest of the land class.

**Table 2.** Natural development scenario cost matrix.

| Land Use Type | Cultivated Land | Forest | Grassland | Water | Construction Land | Unused Land |
|---|---|---|---|---|---|---|
| Cultivated land | 1 | 1 | 0 | 0 | 1 | 0 |
| Forest | 1 | 1 | 0 | 0 | 1 | 0 |
| Grassland | 1 | 1 | 1 | 0 | 1 | 0 |
| Water | 1 | 1 | 0 | 1 | 1 | 0 |
| Construction land | 1 | 0 | 0 | 0 | 1 | 0 |
| Unused land | 1 | 1 | 1 | 1 | 1 | 1 |

Ecological protection scenario (EPS): The EPS's goal is to improve the safeguards for ecological regions including grasslands and forests. According to CP, the development of Chengdu-Chongqing Urban Agglomeration Development Plan, the conversion of wetland to built-up area, pasture, forest, and farming were all strictly regulated (2016–2020) (Table 3). In comparison with natural development, conversion of woodlands and grasslands to the rest of the land class represents the biggest difference. Aside from construction land, other types of land are more easily converted to woodlands and grasslands, and conversion between them is also easier. The probability of woodland and grassland being set to 1 increases as a result.

**Table 3.** Budget matrix for ecological conservation scenarios.

| Type of Land Usage | Cultivated Land | Forest | Grassland | Water | Construction Land | Unused Land |
|---|---|---|---|---|---|---|
| Cultivated land | 1 | 1 | 0 | 0 | 1 | 0 |
| Forest | 1 | 1 | 1 | 1 | 1 | 1 |
| Grassland | 1 | 1 | 1 | 1 | 1 | 1 |
| Water | 1 | 1 | 0 | 1 | 1 | 0 |
| Construction land | 1 | 0 | 0 | 0 | 1 | 0 |
| Unused land | 1 | 1 | 1 | 1 | 1 | 1 |

### 2.3.4. InVEST Model

Using the InVEST model, a regional carbon storage evaluation was conducted. Further, it was investigated if spatial management might successfully stop the loss of region carbon storage [34]. In order to calculate the total carbon storage in a region, the following calculations were made:

$$C_T = \sum_{i=1}^{n} C_{i\_T} = \sum_{i}^{n} A_i \times \left( C_{i\_above} + C_{i\_below} + C_{i\_dead} + C_{i\_soil} \right) \tag{6}$$

where $C_T$ stands for region net carbon storage, $C_{i\_T}$ for i land-use type's carbon storage, $A_i$ for i land-use type's area, and $C_{i\_above}$, $C_{i\_below}$, and $C_{i\_dead}$ but instead $C_{i\_soil}$ for i land-use type's above-ground, below-ground, dead organic matter, and soil carbon densities, respectively.

The carbon density of various land-use types is the fundamental element of the InVEST model, and it is based on previous research findings that have been modified in accordance with the characteristics of the Chengyu Cities Group (Table 4).

**Table 4.** Carbon density based on land-use/cover type included in the InVEST model (t/ha).

| Land-Use Type | Aboveground Carbon Density | Underground Carbon Density | Density of Soil Carbon | Carbon Density of Dead Organic Materials | Sources |
|---|---|---|---|---|---|
| Cultivated land | 38.70 | 80.70 | 92.90 | 1.00 | [37–39] |
| Forest | 55.56 | 144.87 | 206.45 | 3.50 | [39–41] |
| Grassland | 29.30 | 52.90 | 135.00 | 1.00 | [37–40] |
| Water | 21.40 | 73.10 | 113.00 | 1.00 | [41,42] |
| Construction land | 3.30 | 87.30 | 115.30 | 0 | [42,43] |
| Unused land | 22.60 | 136.90 | 171.80 | 0 | [38,42] |

## 3. Results

### 3.1. LUCC Dynamics during 2000–2020

From 2000 to 2020, land use in the Chengdu-Chongqing urban agglomeration was dominated by cultivated land, accounting for more than 57% of the total land area of the urban agglomeration. Woodland occupied more than 29% of the total land area; however, the areas of grassland, water, construction land, and unused land were relatively small, accounting for only 10% of the total area of urban agglomeration land (Table 5). During the past 20 years, land use has changed in different ways, among which the largest change reflects the area of construction land. Increases in unused land, water, building, and forests, are in the increments of 1473, 495, 4393, and 96 km$^2$, respectively. The largest percentage increase occurred for construction land, with an increase of 58.82%. The areas of arable land and grassland decreased by 3609 and 2848 km$^2$, respectively. From 2000 to 2010, under the influence of regional development orientation, continuous urbanization led to the rapid expansion of urban and rural construction land, whereas cultivated land gradually decreased. In addition, the policy of "returning farmland to forest" piloted in the Chengdu-Chongqing region restored forestland area, which was another important reason for the decreasing area of cultivated land. From 2010 to 2020, with further urbanization, urban agglomerations become increasingly large, resulting in a further decrease in the area of cultivated land. Additionally, the expansion of the "returning farmland to forest" project encouraged the ongoing expansion of the forest, while the arable area continued to decrease.

**Table 5.** Area and percentage of the study area's various land-use classifications through 2000 to 2020.

| Land Use Type | 2000 | | 2010 | | 2020 | | Area Change (km²) |
|---|---|---|---|---|---|---|---|
| | Area (km²) | Percentage (%) | Area (km²) | Percentage (%) | Area (km²) | Percentage (%) | |
| Cultivated land | 122,591 | 58.84 | 121,014 | 58.08 | 118,982 | 57.10 | −3609 |
| Forest | 60,696 | 29.13 | 61,812 | 29.66 | 62,169 | 29.84 | 1473 |
| Grassland | 18,944 | 9.09 | 17,021 | 8.17 | 16,096 | 7.73 | −2848 |
| Water | 2839 | 1.36 | 3080 | 1.48 | 3334 | 1.60 | 495 |
| Construction land | 3076 | 1.48 | 5120 | 2.46 | 7469 | 3.58 | 4393 |
| Unused land | 211 | 0.10 | 310 | 0.15 | 307 | 0.15 | 96 |

### 3.2. Analysis of Prediction Results of Various Land Use Situations

In an urban agglomeration, agricultural area, forest areas, grass, water area, and unoccupied land are all expected to shrink by 361, 599, 438, 488, and 20 km² by 2030, respectively, under the natural outcome measurement (NDS in Figure 3). Conversely, construction land area has a projected increase of 25.52%. According to the direction of land-use area transfer (Figure 3), arable land, grassland, and water area will mainly be converted to construction land, whereas forest land and unused land will be evenly transferred to other land types. Although the change range of construction land is the largest, it seldom changes to other land types, and its increase mainly results from the transfer of large areas of cultivated land.

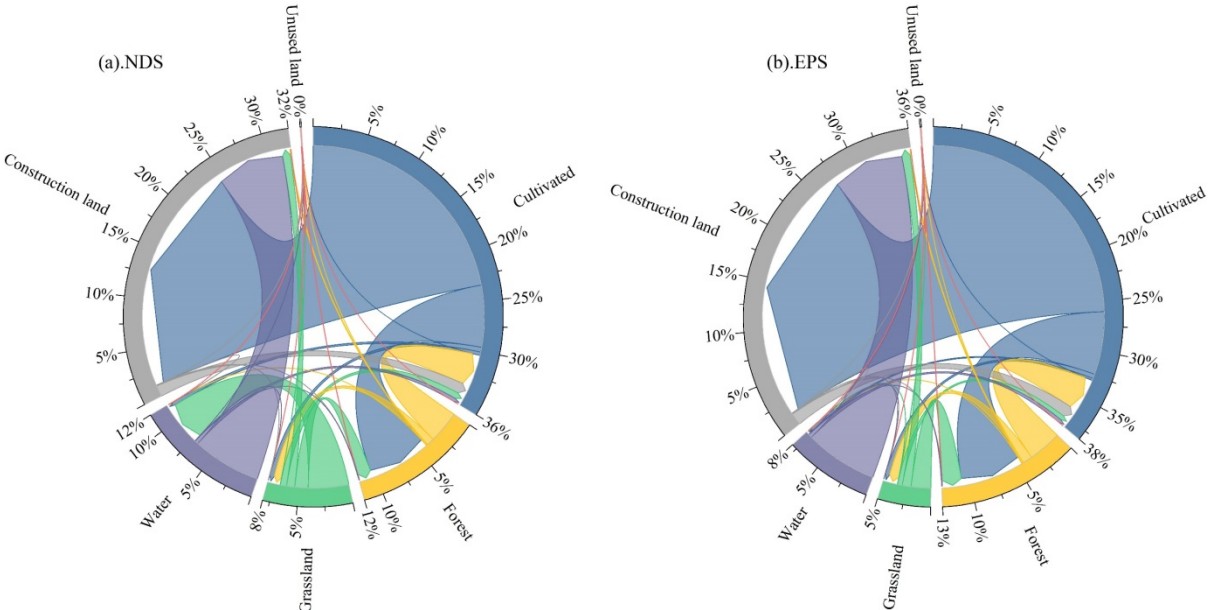

**Figure 3.** Chord diagram of land-use transfer.

Compared with 2020, under the ecological protection scenario, woodland, grassland, and construction land will continue to increase by 101, 345, and 1906, respectively (EPS in Figure 3), While the amount of water, agricultural lands, and undeveloped land will all fall by 2100, 232, and 20, respectively. According to Figure 3, most cultivated land is converted to forest and construction land, most forest land is changed to farmland and grasslands, most grassland is transformed to forestry land and water, and any unused land is evenly distributed to various land types. Relative to the natural development scenario, the change trend of cultivated land, forest land, grassland, and water area under the ecological protection scenario undergoes great changes (Table 6). This primarily results

from the use of land for construction to meet the ecological security patterns of urban agglomeration in the near future. The proportion of forest land, grassland, and water area in the total area of urban agglomeration increased significantly from −0.96%, −2.72%, and −14.64% to 0.16%, 2.14%, and −6.96%, respectively.

**Table 6.** Area of each region in 2030 under the concept of environmental preservation and natural development, and its ratio to 2020.

| Land Use Type | 2020 | | 2030 | | | | Change from 2020 to 2030 | | | |
|---|---|---|---|---|---|---|---|---|---|---|
| | | | NDS | | EPS | | NDS | | EPS | |
| | Area (km²) | Proportion (%) | Area (km²) | Proportion (%) | Area (km²) | Proportion (%) | Area (km²) | Rate (%) | Area (km²) | Rate (%) |
| Cultivated land | 118,982 | 57.10 | 118,621 | 56.93 | 116,882 | 56.10 | −361 | −0.30 | −2100 | −1.77 |
| Forest | 62,169 | 29.84 | 61,570 | 29.55 | 62,270 | 29.89 | −599 | −0.96 | 101 | 0.16 |
| Grassland | 16,096 | 7.73 | 15,658 | 7.52 | 16,441 | 7.89 | −438 | −2.72 | 345 | 2.14 |
| Water | 3334 | 1.60 | 2846 | 1.37 | 3102 | 1.49 | −488 | −14.64 | −232 | −6.96 |
| Construction land | 7469 | 3.58 | 9375 | 4.50 | 9375 | 4.50 | 1906 | 25.52 | 1906 | 25.52 |
| Unused land | 307 | 0.15 | 287 | 0.13 | 287 | 0.13 | −20 | −6.52 | −20 | −6.52 |

### 3.3. Accuracy Verification and Driving Factor Contribution Analysis

Land-use data from 2010 and 2020 were used as examples to simulate changes in land use based on Markov's predictions for every land-use level in 2020. The results were then compared to the actual values for 2020 to assess the PLUS model's simulation accuracy (Figure 4). The method was employed to determine the reliability overall and the kappa coefficient. Values and the kappa coefficient that are near 1 denote simulation accuracy that is higher. The simulated performance of the model reaches a sufficient level in statistical significance whenever the kappa coefficient is higher than 0.75 [44]. The accuracy of kappa was confirmed to be 0.83.

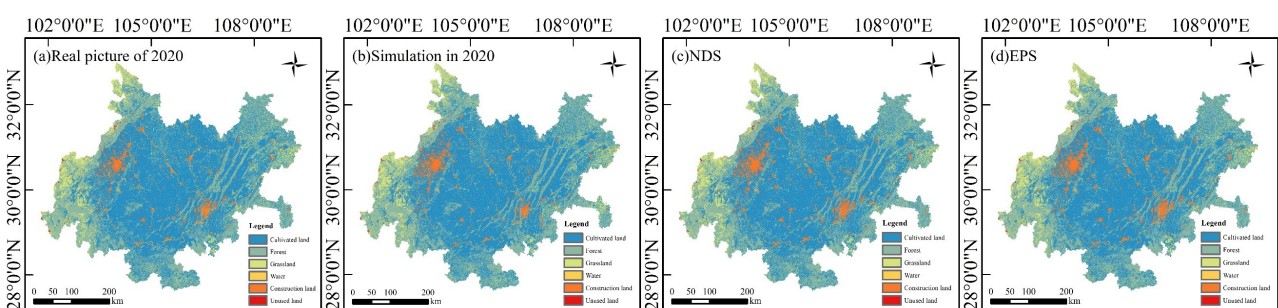

**Figure 4.** Comparison of simulation for 2020 and predictions of two scenarios in 2030.

According to the historical development trend between 2000 and 2020, predictions were made for the contribution ranking of the influencing factors of various land-use probabilities over the next decade (Figure 5). It is obvious from the figure that DEM has the greatest impact on cultivated land expansion, whereas the degree of contribution of other factors is not significantly different. When it comes to the contribution of forest land, the slope factor ranks highest among the fifteen selected driving factors. In addition, DEM has a strong contribution to grassland and water area, indicating that natural environmental factors play an important role. The degrees of contribution of various factors in construction land showed a ladder type. Population factors and DEM factors had the least influence on the unutilized land protrusion.

### 3.4. Changes of Carbon Storage between 2000 and 2030

The InVEST model was used to calculate carbon storage in the Chengdu-Chongqing urban agglomeration for 2000, 2010, and 2020. In order to simulate and forecast land use outcomes in 2030 and to forecast carbon storage capacity within gradual progression and ecological preservation scenarios, this was integrated with the PLUS model. In 2000, 2010, and 2020, the carbon storage of the Chengdu-Chongqing urban agglomeration was $5648.610 \times 10^6$ t, $5669.267 \times 10^6$ t, and $5673.100 \times 10^6$ t, respectively, showing a continuous upward trend with an overall increase of $24.490 \times 10^6$ t and an average annual increase of $1.225 \times 10^6$ t. From 2000 to 2010, carbon storage in the urban agglomeration increased significantly, with an added value of $20.657 \times 10^6$ t, which is an increase of 0.37%. In contrast, from 2010 to 2020, carbon storage in the urban agglomeration increased slightly, with an increment of $3.833 \times 10^6$ t, which is an increase of 0.07%.

In the case of natural development, the carbon storage of the agglomeration in 2030 is predicted to be $5623.099 \times 10^6$ t, a decrease of $50.001 \times 10^6$ t compared with 2020 and reflecting an average annual decrease of $5.0001 \times 10^6$ t. In contrast, under the ecological protection scenario, the carbon storage of the agglomeration in 2030 is predicted to be $5623.347 \times 10^6$ t. The corresponding average annual decrease of $4.9753 \times 10^6$ t indicates that the carbon storage deceleration is small. Under the ecological protection policy, which improves the effectiveness of regional ecological protection and carbon sequestration effects achieved in the Chengdu-Chongqing urban agglomeration. During 2020–2030, compared with the two typical development scenarios, the carbon storage of urban agglomerations under ecological protection measures that restrict the transfer of forest land and grassland to other land types tends to be more stable, avoiding a rapid decline.

Regarding the spatial distribution and evolution of carbon storage (Figure 6), carbon storage in the northwest region with Chengdu as the core of the urban agglomeration decreased slightly between 2000 and 2020 by $0.095 \times 10^6$ t. On the contrary, the southeast region, with Chongqing as the core, exhibits a large increase in carbon storage by $22.722 \times 10^6$ t. From 2020 to 2030, carbon storage of all cities in the Chengdu-Chongqing urban agglomeration decreased under the natural development scenario. Compared with the previous two decades, the northwest region with Chengdu as the core is still the region with the largest reduction in carbon storage, decreasing by $27.923 \times 10^6$ t, accounting for 55.84% of the total reduction. Secondly, the carbon storage in the southeast region with Chongqing as the core exhibited a smaller decreased $22.078 \times 10^6$ t, accounting for 44.16% of the total reduction. Under the ecological protection scenario, the northwest region with Chengdu as the core is still the city with the largest reduction of carbon storage at $27.840 \times 10^6$ t, which corresponds to 99.70% of the natural development scenario, reflecting the effectiveness of ecological protection. The carbon storage of the southeast region with Chongqing as the core also decreased slightly, further reflecting the necessity of ecological protection.

### 3.5. Characteristics of Change in Carbon Storage Caused by Land Type Conversion

Due to area transfer and carbon density differences among different land types, the corresponding effects of change and transformation on carbon storage are different. Due to the change from a single land type to several land types between 2000 and 2020, the Chengdu-Chongqing urban agglomeration lost $25.447 \times 10^6$ t of carbon storage. Quantitative conversion of cultivated land to construction land together with the conversion of forest land and grassland to cultivated land and construction land leads to decreased carbon storage in soil and vegetation. Because water area has a lower carbon density than other land types, converting it to another type of land can help create a carbon sink, which increases the amount of carbon that can be stored overall in the urban agglomeration. During the past 20 years, the transfer of cultivated land to other land types resulted in a reduction of carbon storage of $114.940 \times 10^6$ t, with cultivated land mainly being converted into forest land, grassland, and construction land. Increasing conversion of forest land to other land types also increased, and the corresponding added value was $148.844 \times 10^6$ t. The carbon storage of grassland correlated with the area change observed over the last

20 years; i.e., with a decrease in area, its carbon storage also decreased, and the conversion between different land types decreased by $57.241 \times 10^6$ t. Because the carbon density of water areas is low and because its area does not change greatly, the carbon storage value of water areas does not change significantly between land type conversions. Due to its strong expansion, construction land increased continuously during the past 20 years and was mainly converted to arable land, forest land, and grassland. The unused land showed an overall trend of fluctuating growth, resulting in a small increase of carbon storage between land conversion, with an added value of $0.360 \times 10^6$ t.

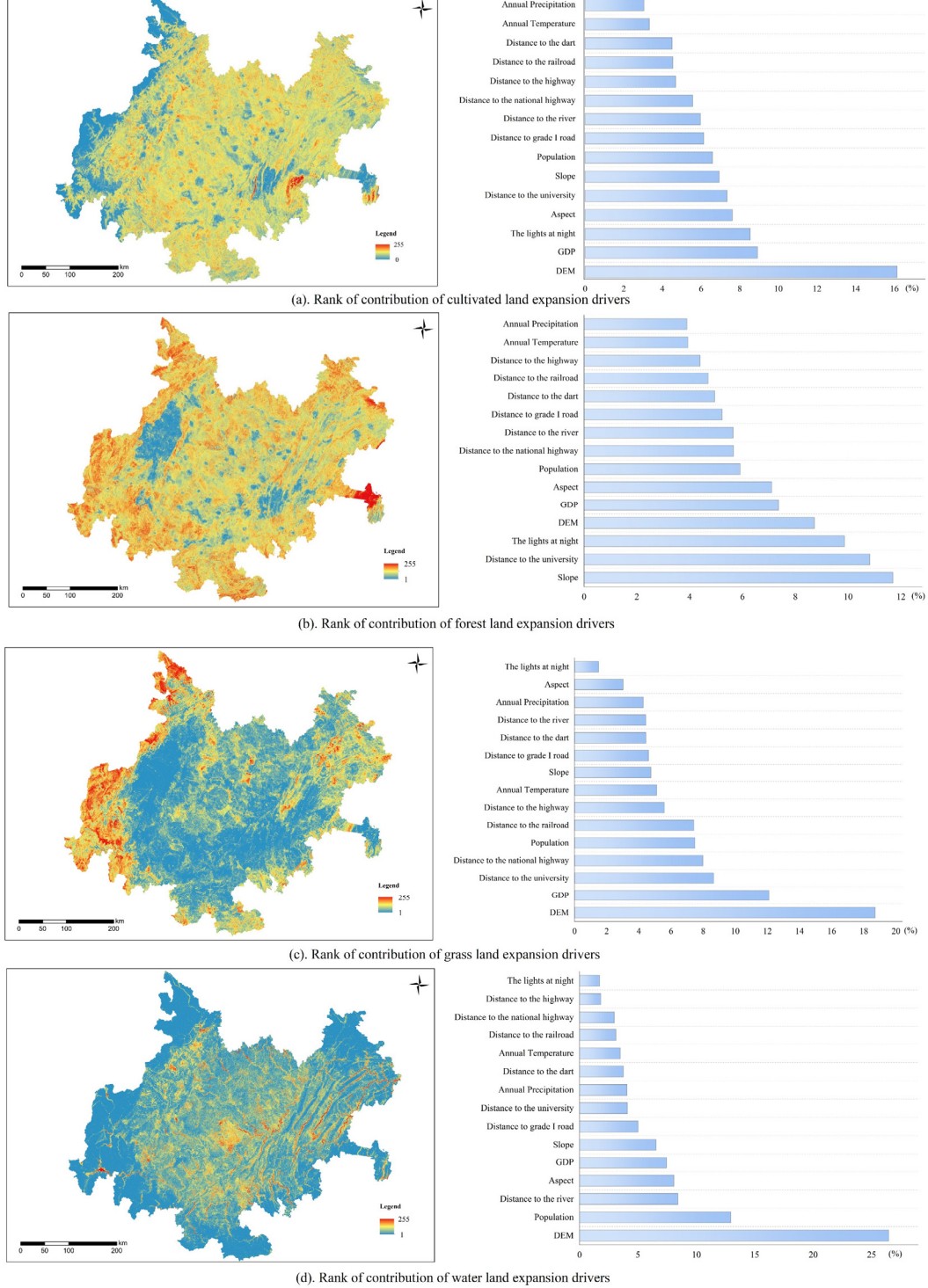

(a). Rank of contribution of cultivated land expansion drivers

(b). Rank of contribution of forest land expansion drivers

(c). Rank of contribution of grass land expansion drivers

(d). Rank of contribution of water land expansion drivers

**Figure 5.** *Cont.*

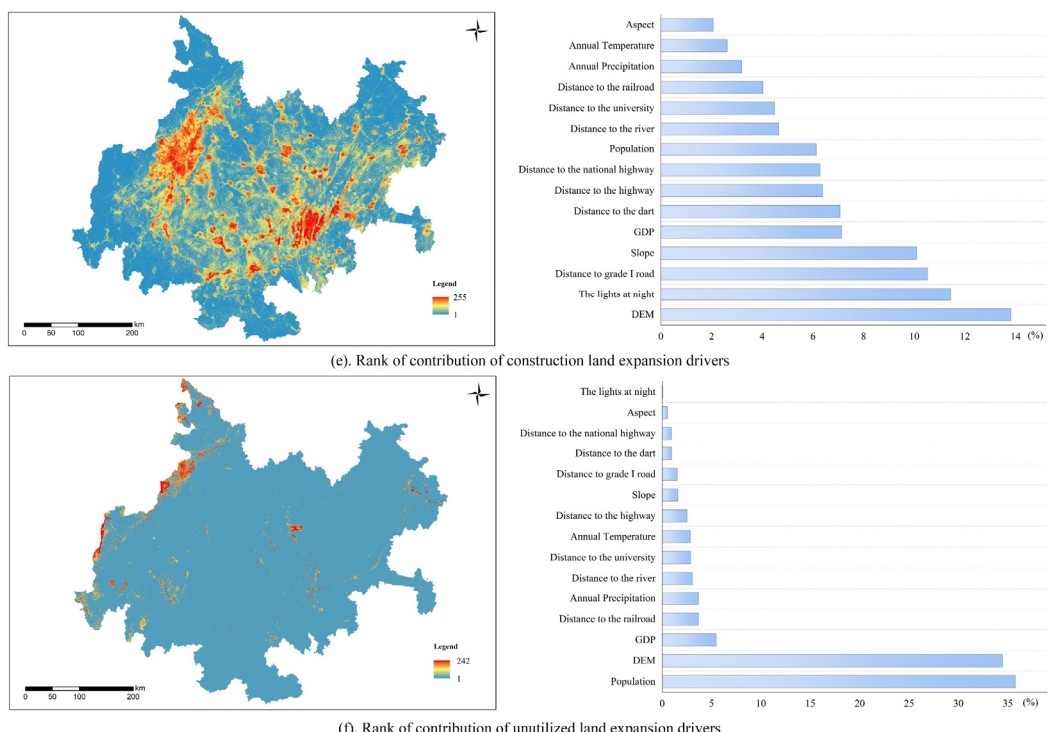

(e). Rank of contribution of construction land expansion drivers

(f). Rank of contribution of unutilized land expansion drivers

**Figure 5.** Ranking of various land-use probabilities and their driving factors.

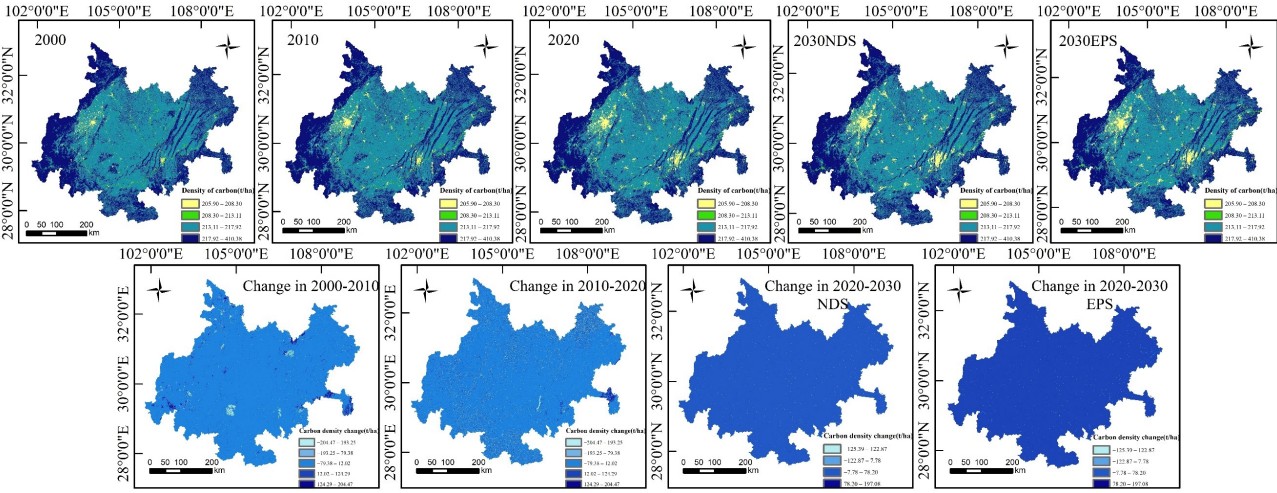

**Figure 6.** Carbon storage and its changes over different periods.

Compared with 2020 (Table 7), the carbon storage of the Chengdu-Chongqing urban agglomeration decreased by $2.955 \times 10^6$ t under the natural scenario by 2030 and significantly increased by $393.057 \times 10^6$ t under the ecological protection scenario. The main reason for this is the different transfer probability of cultivated land, forest land, grassland, and water area. In addition, guided by ecological protection, the conversion of other land types is restricted, and the transfer area to construction land decreases, resulting in increased carbon storage. Under the two tested scenarios, the carbon storage of forest land is the most significant. Despite the fact that each scenario indicated an upward trend, the carbon storage of forestland increased significantly under ecological protection scenario. First, there is a decline in the transformation of forested areas to agricultural land. Second, forestland controls the transfer of construction land and unused land, achieving the goal of regional carbon stability and reflecting the effectiveness and necessity of ecological

protection policies. The change of carbon storage in grassland and water area was not obvious. The ability to store more carbon is significantly increased by converting water areas, building sites, and unused land into agricultural land, forest areas, and grassland. Overall, the conversion of various land types mainly results in increased construction land, which will prevent the metropolitan agglomeration from growing its carbon store in the foreseeable.

**Table 7.** Under scenarios of natural progression and ecological protection in 2020–2030, land type conversion will influence the amount of carbon stored in the atmosphere.

| Land Use Type | | Area (km$^2$) | | Change in Carbon Stock ($\times 10^6$ t) | | Total ($\times 10^6$ t) | |
|---|---|---|---|---|---|---|---|
| Converted from | Converted to | NDS Natural Development Scenario | EPS Ecological Protection Scenario | NDS Natural Development Scenario | EPS Ecological Protection Scenario | NDS Natural Development Scenario | EPS Ecological Protection Scenario |
| Cultivated land | Forest | 411.76 | 380.44 | −8.115 | −7.498 | −6.970 | −6.341 |
| | Grassland | 32.08 | 26.89 | −0.016 | −0.013 | | |
| | water | 22.48 | 12.75 | 0.011 | 0.006 | | |
| | Construction land | 1557.73 | 1576.12 | 1.153 | 1.166 | | |
| | Unused land | 0.23 | 0.24 | −0.003 | −0.003 | | |
| Forest | Cultivated land | 215.28 | 203.13 | 4.243 | 400.321 | 5.395 | 401.655 |
| | Grassland | 50.17 | 59.49 | 0.964 | 1.143 | | |
| | water | 1.16 | 1.09 | 0.023 | 0.022 | | |
| | Construction land | 7.68 | 7.87 | 0.157 | 0.161 | | |
| | Unused land | 1.00 | 1.01 | 0.008 | 0.008 | | |
| Grassland | Cultivated land | 42.01 | 28.14 | 0.021 | 0.014 | −1.362 | −2.230 |
| | Forest | 88.56 | 119.67 | −1.702 | −2.300 | | |
| | water | 270.12 | 0.70 | 0.262 | 0.001 | | |
| | Construction land | 54.92 | 53.23 | 0.068 | 0.065 | | |
| | Unused land | 0.92 | 0.87 | −0.010 | −0.010 | | |
| Water | Cultivated land | 23.16 | 21.05 | −0.011 | −0.010 | 0.071 | 0.067 |
| | Forest | 1.49 | 1.49 | −0.030 | −0.030 | | |
| | Grassland | 1.00 | 0.71 | −0.001 | −0.001 | | |
| | Construction land | 458.90 | 442.21 | 0.119 | 0.115 | | |
| | Unused land | 0.53 | 0.54 | −0.006 | −0.007 | | |
| Construction land | Cultivated land | 72.96 | 72.88 | −0.054 | −0.054 | −0.118 | −0.121 |
| | Forest | 3.02 | 3.16 | −0.062 | −0.065 | | |
| | Grassland | 0.76 | 0.75 | −0.001 | −0.001 | | |
| | water | 2.38 | 2.45 | −0.001 | −0.001 | | |
| | Unused land | 0.06 | 0.04 | −0.001 | 0.000 | | |
| Unused land | Cultivated land | 0.19 | 0.21 | 0.002 | 0.002 | 0.030 | 0.026 |
| | Forest | 1.81 | 2.08 | −0.014 | −0.016 | | |
| | Grassland | 0.89 | 0.85 | 0.010 | 0.010 | | |
| | water | 1.66 | 1.46 | 0.020 | 0.018 | | |
| | Construction land | 0.89 | 0.99 | 0.011 | 0.012 | | |
| Total ($\times 10^6$ t) | | | | | | −2.955 | 393.057 |

## 4. Discussion

### 4.1. PLUS Analysis of Model Uncertainty

Currently, the majority of research on LUCC-related alterations to ecosystem carbon cycles is based on model simulations. Due to its complexity, LUCC can affect the energy flow in the ecosystem, but the existing models are hindered by uncertainties in simulating changes in the ecosystem carbon cycle caused by LUCC [34,45]. The reliability of prospective land-use change modeling scenarios largely determines the accuracy of modeling findings.

The driving factors selected for the PLUS model simulation used in this paper are terrain conditions, climate environment, social economy, and transportation accessibility, including fifteen factors, such as DEM, slope, aspect, temperature, precipitation, GDP, population, railway networks, and national road networks. These factors result in an accurate simulation of various land types and have different contributions to different

types of land (Figure 4). Nonetheless, the PLUS model still has some limitations. First, in addition to natural factors, cultural factors feature many complex choices, such as cultural concepts, industrial output value, and POI; however, because it is challenging to quantify these elements inside the PLUS model, we do not incorporate them in our simulation method. Secondly, national policy orientation plays an important role in LUCC. Policy factors such as ecological protection red line, permanent basic farmland protection red line, and urban development red line—which all play a very important role in China's territorial space change—are difficult to assign specific values in the simulation process due to their complexity. Therefore, in order to better adapt to real scenarios of future land-use change, it is necessary to consider introducing more influencing factors in subsequent studies.

*4.2. InVEST Model Uncertainty Analysis*

The InVEST model can intuitively determine the effects of different types of conversion on carbon storage. Its results clearly reflect spatial and temporal variations of carbon storage in urban agglomerations and highlight the relationships between different land types, which can provide new ideas for regional development in terms of coordinating economic and ecological aspects. Nonetheless, it is important to note that the InVEST model makes more estimates for large-scale land changes based on established available carbon density values. In the carbon module, the change of carbon storage value due to vegetation growth and the internal structure of land use are ignored, resulting in errors in the change of spatial patterns of carbon storage and leading to uncertainty in the results [27]. In addition, although the carbon density values obtained from existing studies are close to those in the study area, these values may be influenced by human activities and changes in the natural environment. Therefore, the carbon density value also has a certain degree of uncertainty. Finally, while the carbon module considers differences in carbon density between different land-use types, it ignores differences in carbon sinks related to land-use types and the age organization of vegetation, which hinders the simulation of the estimation of the spatial pattern of carbon storage services. Therefore, in the study of future urban agglomeration, it is necessary to strengthen and verify the timeliness of data acquisition of carbon density values, carry out localized calibration, conduct field measurements of core indicators, and accurately estimate changes in regional carbon storage and based the on scientific and reasonable assumptions in order to better maintain the carbon balance of the regional ecosystem.

*4.3. Advantages and Limitations of the Linkage Model*

The Link PLUS and InVEST models have broad applications for guiding ecosystem services. The PLUS model's LEAS module extracts the growth of different types of property between the two steps for land-use change, which collects samples from the growing section. In order to investigate the variables of development probability related to each land-use type and assess the impact of each factor driving on land-use type expansion, the advancement and pushing factors of each land-use type are then paired using the random forest method [46]. This allows land-use change simulations combined with the InVEST model to be used as a means for studying future changes in regional carbon storage spatial patterns within urban agglomerations.

Although linkage models can effectively simulate the effects of ecosystems on carbon storage over short time scales, their application on longer time scales faces several limitations. The regional climate of the Chengdu-Chongqing urban agglomeration is humid all year, causing vegetation and soil carbon density to change constantly [47]. Consequently, the consequences of climate change could be disregarded whenever relational models are utilized for long-term projections. In addition, the original spatial resolution of LUCC data used was 30 m × 30 m. To ensure consistency, all spatial data were resampled to a grid of 100 m × 100 m. In future studies, data accuracy could be further improved to ensure the validity of simulation sampling and carbon storage measurements.

*4.4. Spatial Structure of Urban Agglomerations and Carbon Storage*

Urban agglomeration is a highly integrated urban complex with compact spatial organization and close economic ties formed by different levels of cities relying on transportation and communication and other infrastructure networks in a specific geographical area [48,49]. Agglomerations of urban space are dependent on land as a space carrier for social, economic, and ecological activities [50]. The land is also one of the most essential spatial attributes of urban development. Further, land-use type and its cover change represent the concrete expression of land as well as the main manifestation of urban agglomeration structure. Therefore, LUCC, which is crucial to the carbon cycle in terrestrial ecosystems, is at the heart of the urban agglomeration-carbon storage relationship.

In this study, we analyze the changes in carbon reserves caused by LUCC in Chengdu-Chongqing urban agglomerations, based on the relationship between urban spatial structure and carbon reserves. There was a significant change in cultivated land from 2000 to 2020. By converting this land type to another, the total area decreases, resulting in the largest decrease in carbon reserves. By contrast, forest land increases its carbon reserves as its transfer area increases. Other land types are similarly affected. In accordance with the historical development, the relationship between the spatial structure and carbon reserves of the natural scenario will remain the same in ten years. The total amount of carbon stored also increases as forest land is converted to grassland in the ecological scenario. There is no doubt that changes in land use will affect carbon reserves over time. There is a direct correlation between urban agglomeration's spatial structure and carbon storage. This relationship is traceable. According to Nicodemus Nyamari [51] and Cai [7], carbon reserves have changed in Kenya and China's Yangtze River Delta due to LUCC. In general, it can be observed that urban agglomerations and carbon reserves have a close relationship, and any changes in one will inevitably affect the other.

*4.5. Development Strategy for Urban Agglomeration and Carbon Storage*

In recent years, with the continuous development of urban agglomerations, contradictions of land use caused by urban expansion have become increasingly common. Continuously changing the land-use type can have a negative impact on the carbon sink of terrestrial ecosystems [52]. Consequently, China's territorial space planning must advance in order to achieve regional economic development goals while ensuring ecological protection in urban agglomerations. First, it is necessary to strictly abide by the "three red lines for protection" guided by national policies and appropriately control the increase of land used for construction. An example is Chengdu-Chongqing's urban agglomeration. Construction land in Chengdu and Chongqing as well as their surrounding large cities should be developed at a reduced pace. It is anticipated that small- and medium-sized cities will grow moderately because they are not occupying arable land. Further, the city needs to renovate the new construction space in order to tap into the potential of low-efficiency land and increase urban public and ecological space. Increasing construction land intensity is crucial to achieving limited growth and spatial transfer incentives in small towns and villages within urban agglomerations. Second, the ratio of forestland to grassland area should be increased to strengthen the ecological protection of high carbon density regions [53,54]. Therefore, it is imperative that the Chengdu-Chongqing urban agglomeration not only increase forest cover through afforestation in large areas but also optimize the urban vegetation structure. The objective is to guide the sustainable development of urban forest land and grasslands and to realize a harmonious coexistence pattern of life ecology. Enhance the total carbon storage capacity of the urban agglomeration and create an urban ecosystem that is healthy and stable as well as a harmonious living environment. Finally, it is imperative to focus on the complex function of land uses in order to complete the transformation from single land type to a production-life-ecological complex function. It is necessary to increase participation in ecological preservation, promote ideal land layouts, minimize carbon dioxide emissions, and enhance ecological efficiency thru the utilization of resources and large-scale land management. For urban agglomerations and metropolitan

areas in other regions, this has certain reference value. Building an ecological security pattern and promoting an effective carbon cycle are the meanings of national development from the perspective of urban agglomeration.

*4.6. Contribution to Research*

Several contributions make this study different from others. On the one hand, this article quotes the latest simulation prediction model, PLUS, which has characteristics that the previous forecast model lacks. Various types of land can be better understood through this method. In addition, it contains a new multi-type seed growth mechanism that can simulate changes in plaque-level and land-level changes for a variety of land types. Moreover, it is coupled with a variety of target algorithms, enabling better planning decisions to be made. A FLUS model was used in previous research by Zuo et al. [55] to simulate land-use changes in 2020 in Chongqing. Additionally, Zhang et al. [56] simulated mainland China's ecosystem value using the FLUS model. In their research, they only needed to extract the first phase of the land in order to use the data for training, based on the probability of emergence and land competition. There is a lack of time concepts in this method as well as the ability to dig the changes in land use compared to the PLUS model.

On the other hand, Chengdu-Chongqing urban agglomeration is the largest urban group in southwestern China. In national strategic development, the Chengdu-Chongqing urban agglomeration plays a significant role due to their geographical location and historical significance. Carbon reserves in this area not only meet the needs of local ecological development but also get closer to the national dual carbon strategic planning goals. Data integration is also performed from a macro perspective, and different models can be used to better position Chengdu-Chongqing urban development. For the remaining urban agglomerations, this is of more guiding significance. Previous simulation prediction research using the PLUS model focused mostly on middle and small regions, similar to cities or specific places. In their analyses of the Hanzhong ecosystem and the southeast coastal protective forest ecosystem, Yang [57] and Bao [58] used the PLUS models. Study area is small and does not have universality for urban areas, urban agglomerations, or other regions. The main contributions of this study are therefore the two aspects above. The study presents multiple suggestions to improve the reference value of national strategic planning and regional development based on the results of the scenario simulation.

**5. Conclusions**

This study made projections for the carbon storage of something like the Chengdu-Chongqing metropolitan agglomeration in 2030 using the PLUS and InVEST models. These are its conclusions:

(1) Land use in the Chengdu-Chongqing urban agglomeration has changed significantly between 2000 and 2020, primarily due to a continuous increase of forest land area, water area, construction land area, and unused land area, together with a decrease of cropland and grassland areas. The driving force behind this change mainly comes from urbanization and the implementation of the "returning farmland to forest" policy. Carbon storage in the urban agglomeration has increased by $24.490 \times 10^6$ t in the past 20 years.

(2) In comparison, the accuracy of kappa is 0.83. According to the historical development trends from 2000 to 2020, the contribution of the probability impact factors of regional expansion have been calculated and ranked. The DEM exerts a significant influence, but other factors also contribute differently in specific situations.

(3) From 2020 to 2030, the cultivated lands, forests, grasslands, water areas, and unused lands in Chengdu-Chongqing will decline continuously under the natural development scenario. The area of construction land will continue to grow. The urban agglomeration's carbon storage will decrease from $5673.100 \times 10^6$ t in 2020 to $5623.099 \times 10^6$ t in 2030, i.e., a total decrease of $50.001 \times 10^6$ t.

(4) In the scenario of ecological preservation, crop land, water area, and unoccupied land will all decrease, while woods, grassland, and building land would all continue to grow. In this scenario, the urban agglomeration's carbon storage in 2020 will decrease from $5673.100 \times 10^6$ t to $5623.347 \times 10^6$ t in 2030, i.e., a total decrease of $49.753 \times 10^6$ t.

(5) Carbon storage under the ecological protection scenario can be reduced by $0.248 \times 10^6$ t relative to the natural development model. This slower reduction rate is conducive to the stabilization of carbon sinks. Under the ecological protection scenario, carbon storage in northwest China with Chengdu as its core decreased by $27.840 \times 10^6$ t, i.e., 99.70% of the natural development scenario. Carbon storage in southeast China, with Chongqing as its core, also declined slightly.

**Author Contributions:** Conceptualization, C.W. (Chaoyue Wang) and X.G.; methodology, C.W. (Chaoyue Wang), X.G. and T.L.; validation, C.W. (Chaoyue Wang), X.G. and T.L.; investigation, L.X., C.L. and C.W. (Chunbo Wang); writing—original draft, C.W. (Chaoyue Wang); writing—review and editing, C.W. (Chaoyue Wang), X.G. and T.L.; software, C.W. (Chaoyue Wang) All authors have read and agreed to the published version of the manuscript.

**Funding:** The Chongqing Education Commission's Humanities and Sociology Research Program (No. 21SKGH432) and the National Sociology Foundation of China (No. 21BMZ141) provided funding for this work. Project supported by China National Scholarship Foundation.

**Institutional Review Board Statement:** Not applicable.

**Informed Consent Statement:** Not applicable.

**Data Availability Statement:** Publicly available datasets were analyzed in this study. This data can be found here: Geospatial Data Cloud Website (http://www.gscloud.cn, accessed on 6 June 2021); RESDC (http://www.resdc.cn/, accessed on 12 June 2021); OpenStreetMap (https://www.openstreetmap.org/, accessed on 2 July 2021); World Clim (https://www.worldclim.org/, accessed on 16 August 2021).

**Acknowledgments:** We value the reviewers' critical and helpful criticism and recommendations, which boosted this manuscript's quality.

**Conflicts of Interest:** The authors say they have no competing interests. The study's design, data collection, analysis, or interpretation; the preparation of the paper; or the choice to publish the findings were all made independently of the funding sponsors.

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
