# Peer review of "Plus-InVEST Study of the Chengdu-Chongqing Urban Agglomeration’s Land-Use Change and Carbon Storage"

_land, doi:10.3390/land11101617_

Round 1

Reviewer 1 Report

General Comments:

I read the article and identified a quality production, at the level of your journal. There is an adequate number of bibliographic references (51), and in general they are recent studies, nothing very old. The objectives are clear and concise, the methodology is adequate for the proposal, the mathematical formulations are ok, the area of study was well defined, and the Chord diagram is a robust graph and extremely rich in information, which is usually difficult to find in the articles. The simulations were well defined and the figures are of good quality. I did not identify methodological errors or open questions about the proposal. For this reason, my opinion as a reviewer is to approve the article.

The article provides a good overview about storing carbon with in Chengdu-Chongqing urban agglomeration. It is within the scope of the journal, well written and well structured. In my opinion, it's great to publish.

Author Response

We appreciate your acknowledgement of our research results and your review of every part of our article. This recognition will motivate us to do better research. Thank you very much.

Reviewer 2 Report

Dear Reviewers,

I have added some comments regarding the reviews of your manuscript. Please, address the comments before the manuscript is ready for publication.

Line 22:  Compared to2020.

Line 57 – 58: check the sentence if both 'future' in this line can be sensible in this specific sentence.

Line 60: Begin this sentence as: The model PLUS includes . . .

Line 76 – 74: This sentence is either incomplete or not worth adding here. It does not give any sense to the previous or succeeding sentences. Check it.

Line 78 – 79: This sentence is incomplete; please make your remark clear. Here, you tried to state the objective of the study. You can modify this sentence as: The objective of the study is to explore. . .

Line 101: Spell out abbreviations (TM, ETM, and OLI) at their appearances.

Line 104: Spell out abbreviation (RESDC) at its first appearances.

Line 149: The figure did NOT clearly indicates which panels belongs to Chengdu or Chongqing. Please name each panel to which urban belongs to.

Line 353: Spell out the abbreviations in the table of the second row in the footnote to make clear for readers what these abbreviations are.

With best,

The reviewer

Author Response

Thank you for acknowledging and commenting on our research. We have prepared a response letter based on your comments. Please see the attachment.

Reviewer 3 Report

The paper addresses the issue of carbon storage capcity within specific region and its dynamic with regard to the past and predicted land use changes. the paper is well designed and well written. The methodological approach is rather good and well justified, with some minor drawbacks as mentioned in the attached manuscript. The editorial and graphic side of the paper are ok but due to the big volume of presented data, sometimes the illustrations are hardly readible. The resources that are used within the text are both actual and well connected to the subject of the paper. Perhaps, the discussion section is missing some more references in order to put the results of the research versus similar studies, especially while the methodological issues are concerned. Some additional remarks are made within the attached manuscript of the paper.

Author Response

(The authors gave the same response as above.)
